# Harnessing Code Interpreters for Enhanced Predictive Modeling: A Case Study on High-Density Lipoprotein Level Estimation in Romanian Diabetic Patients

**DOI:** 10.3390/jpm13101466

**Published:** 2023-10-06

**Authors:** Maitham Abdallah Albajy, Maria Mernea, Alexandra Mihaila, Cristian-Emilian Pop, Dan Florin Mihăilescu

**Affiliations:** 1Department of Anatomy, Animal Physiology and Biophysics, Faculty of Biology, University of Bucharest, 91–95 Splaiul Independenței Str., 050095 Bucharest, Romania; albajy.maitham@s.bio.unibuc.ro (M.A.A.); dan.mihailescu@bio.unibuc.ro (D.F.M.); 2National Center for Occupational Health and Safety, 22 Imam Ali Str., Nasiriyah 64001, Iraq; 3Liberty Medical Center Clinic, Intrarea Zorilor 23 A Str., 077175 Bucharest, Romania; alexandra.mihaila2727@gmail.com; 4Non-Governmental Research Organization Biologic, Schitului 14 Str., 032044 Bucharest, Romania; 5Biometric Psychiatric Genetics Research Unit, Alexandru Obregia Psychiatric Hospital, Șoseaua Berceni 10 Str., 041914 Bucharest, Romania

**Keywords:** diabetes, data analysis, parameters, data modeling, predictors, management strategies

## Abstract

Diabetes is a condition accompanied by the alteration of body parameters, including those related to lipids like triglyceride (TG), low-density lipoproteins (LDLs), and high-density lipoproteins (HDLs). The latter are grouped under the term dyslipidemia and are considered a risk factor for cardiovascular events. In the present work, we analyzed the complex relationships between twelve parameters (disease status, age, sex, body mass index, systolic blood pressure, diastolic blood pressure, TG, HDL, LDL, glucose, HbA1c levels, and disease onset) of patients with diabetes from Romania. An initial prospective analysis showed that HDL is inversely correlated with most of the parameters; therefore, we further analyzed the dependence of HDLs on the other factors. The analysis was conducted with the Code Interpreter plugin of ChatGPT, which was used to build several models from which Random Forest performed best. The principal predictors of HDLs were TG, LDL, and HbA1c levels. Random Forest models were used to model all parameters, showing that blood pressure and HbA1c can be predicted based on the other parameters with the least error, while the less predictable parameters were TG and LDL levels. By conducting the present study using the ChatGPT Code Interpreter, we show that elaborate analysis methods are at hand and easy to apply by researchers with limited computational resources. The insight that can be gained from such an approach, such as what we obtained on HDL level predictors in diabetes, could be relevant for deriving novel management strategies and therapeutic approaches.

## 1. Introduction

Type 2 diabetes mellitus (T2DM) occurs when the body is unable to preserve normal glucose levels due to insulin resistance and β-cell dysfunction [1,2]. T2DM is a complex pathology with a multifactorial etiology. There is a genetic component of the disease, but environmental factors like age, lifestyle, and diet are extremely important [3,4]. Obesity is a major risk factor for T2DM, the body mass index (BMI) being proportional to insulin levels and insulin resistance [4]. An increased calorie intake doubled by a decreased energy consumption, as well as the nutrient composition of food, are also T2DM risk factors. A diet rich in saturated fats was linked to all events that accompany T2DM [1]. Recent studies showed a bidirectional relationship between lipid metabolism and insulin resistance or T2DM, the underlying mechanisms being discussed in [3]. Dyslipidemia is a condition characterized by elevated triglyceride (TG) and low-density lipoprotein (LDL) levels doubled by decreased levels of high-density lipoproteins (HDLs). It is highly prevalent in diabetes [5,6,7] and is a risk factor for cardiovascular diseases [8].

In a previous study, we analyzed the prevalence of metabolic syndrome in 110 T2DM patients from Romania [9]. The metabolic syndrome includes various symptoms related to metabolic dysfunctions, including insulin resistance, obesity, hypertension, and atherogenic dyslipidemia [10]. Several parameters were measured in order to diagnose the metabolic syndrome in these patients, like the levels of glucose, TG, HDL, LDL, BMI, blood pressure, or the level of protein in the urine. In the present study, we continued the analysis of the data in relationship with diabetes. We aimed to identify the parameters best correlated with diabetes and the relationship between them by applying more elaborate data analysis methods. At the same time, we were interested in conducting the analysis with minimum computational expense, and therefore, we used the Code Interpreter feature of ChatGPT [11].

The Code Interpreter plugin allows the upload or download of data, their analysis, and graphical representation using a Python interpreter [12,13]. Even if there are current limitations regarding the analysis of big data like those from the bioinformatics field [13], the ChatGPT Code Interpreter holds the promise of facilitating data analysis even for non-specialists. Our analysis identified HDL as best correlated with all measured parameters, and different models were used to understand HDLs’ relationship to them. Given that diabetes is a risk factor for cardiovascular disease and HDLs present protective vascular effects [14], the insights that we derived could have significant relevance for the treatment of diabetes and for deriving novel management strategies. Also, we showed that a complex data analysis is at hand just by using the Code Interpreter.

## 2. Materials and Methods

### 2.1. Diabetes Dataset

As described in [9], the dataset was acquired cross-sectionally on 110 diabetic and 50 nondiabetic (control, healthy cases) persons that presented for diagnosis or follow-up at The National Institute of Diabetes, Nutrition, and Metabolic Diseases “N.C. Paulescu”, Bucharest, Romania, in the March 2022–November 2022 time period. Almost all diabetic patients were using diabetes treatment consisting of metformin or insulin injections in equal proportions. Supplementarily, the patients also used gliclazide, pioglitazone, or canagliflozin. The patients were prohibited from using any type of medication 24 h prior to sample collection. Twelve parameters were measured, as presented in Table 1.

### 2.2. Data Correlations in Diabetes and Non-Diabetes Datasets

Before using ChatGPT Code Interpreter, the data were analyzed in R environment [15] by descriptive statistics and by correlation analysis [16] between the variables associated with diabetic (Disease = 1) and nondiabetic (Disease = 0) conditions.

### 2.3. Data Analysis

Data analysis was performed with Code Interpreter plugin in ChatGPT [11,12] using the diabetic patients’ dataset (110 values with the parameter Disease = 1). The conversations are given in the Appendix A.

#### 2.3.1. HDL Prediction Based on the Other Variables

HDL levels (dependent variable) were modeled based on the other parameters (independent variables). The following models were considered: Generalized Linear Model (GLM) [17], Lasso Regression [18], Elastic Net [19], Quantile Regression [20], Random Forest [21,22], Gradient Boosting Machine (GBM) [23], Support Vector Machine (SVR) [24], and Neural Network [25]. The 110 subjects were divided into training (88 samples) and test (22 samples) sets. Training of the models involved an initial checking of the preconditions assumed by each model. This was followed by the actual training of the models for which preconditions are met. Additional refinement of the models was achieved by hyperparameter tuning.

##### Preliminary Checks and Preconditions

Preconditions (Table 2) were checked for the above-mentioned models. HDL levels were considered as dependent variables, and the other parameters were considered as independent variables.

The assumptions of linearity and independence were addressed by visual inspection of the scatter plots of residual versus predicted values. The independence and normality of residual assumptions were tested by applying Durbin–Watson statistics [26] and Shapiro–Wilk test [27], respectively. The multicollinearity was tested using the Variance Inflation Factors (VIF) [28]. VIFs were refined by omitting the most inflated variables. The scales of the data were compared in order to determine if data normalization should be applied prior to training the machine learning models.

##### Model Selection and Comparison

The models were applied to our data on diabetes patients, and the performance was estimated as Mean Squared Error (MSE). The performance of the models was improved by tuning and refinement. The model leading to the best MSE was further explored and analyzed.

#### 2.3.2. Random Forest Models of All Variables

Random Forest models were trained for modeling all variables except for disease and onset. Each variable in our dataset was considered dependent, while the rest of the variables were considered independent. Random Forest models were trained for all possible combinations, and the performances of models were compared based on MSE values.

#### 2.3.3. Modeling the Transition from Normal to Elevated HbA1c

The transition from normal to elevated levels of HbA1c was investigated using the Kaplan–Meier survival analysis [29]. The event of interest was defined as the transition from normal (<6.5%) to elevated (≥6.5%) HbA1c levels. The model assumes that “survival” refers to the probability of maintaining HbA1c below the threshold of 6.5%. The variable ‘Onset’ was defined as the time after the initial diagnosis when the event occurred. Patients with HbA1c below the threshold levels were considered as censored.

#### 2.3.4. Software and Tools

As stated by ChatGPT, the analysis was conducted in Python [30], the tools used for statistical analyses and visualizations being pandas [31], numpy [32], scipy [33], statsmodels [34], sklearn [35], matplotlib [36], and seaborn [37] libraries.

## 3. Results and Discussions

### 3.1. Correlation of Analyzed Parameters

An initial analysis of data involved the identification of parameters that are correlated. The analysis was performed separately in the case of diabetic and control patients in order to identify the correlations specific to diabetes (Figure 1). In control patients, most parameters present no association, as supported by the very small correlation coefficients. An exception is the very strong positive correlation between age and systolic blood pressure.

In the case of diabetic patients, there are very strong positive correlations between glucose and HbA1c levels, TG and LDL levels, and glucose or HbA1c levels. Very strong negative correlations are seen between age and HDL levels. Strong positive correlations are seen between age and TG, LDL, glucose, or HbA1c levels and between HDL and other parameters like TG, LDL, glucose, and HbA1c levels. Moderate correlations are seen between Blood Pressure—H and age, BMI, or LDL levels (positive correlation), between age and disease onset (positive correlation), and between Blood Pressure—H and HDL (negative correlation). The other correlations are weak and very weak, like in the case of disease onset and all parameters except for age or in the case of BMI and all parameters except for Blood Pressure—H.

The two plots in Figure 1 show that the data on diabetic patients present many positive and negative correlations that are not seen in the case of control patients. Also, the strong and very strong correlations of variables in diabetic patients suggest that simple linear regressions will not be suited for modeling these parameters.

### 3.2. Model Selection for Predicting HDL Levels

#### 3.2.1. Assumptions and Preconditions Checking

Several models were proposed for modeling HDLs based on the other variables, from simple GLM to complex machine learning approaches. Prior to modeling, the assumptions of linearity (linear relationship between HDL and the independent variables), independence (the observations are independent of each other), homoscedasticity (constant variance across observations), and no multicollinearity (independent variables are not correlated with each other) were checked for the dataset comprising diabetes patients’ parameters.

Linearity between predictors and the dependent variable (HDL) was determined by visual inspection of the corresponding scatter plots. The analysis revealed that some variables show a linear relationship with HDL, while others do not show such a relationship. The homoscedasticity analysis was performed based on the residual plots. These showed a random scattering of variables around the Ox axis (y = 0), suggesting that the assumption is met. The independence of errors was addressed using the Durbin–Watson statistic [26]. The obtained value of 2.35 indicates no autocorrelation of residuals. Additionally, in the context of the GLM model, the normality of error distribution was addressed by the Shapiro–Wilk test. When applying the test, the null hypothesis was that the residuals were normally distributed. The obtained *p*-value of 0.378 suggests that the null hypothesis cannot be rejected; thus, the residuals can be assumed to be normally distributed. These show that the GLM assumptions of linearity, homoscedasticity, independence of errors, and normality of error distribution are reasonably met.

The multicollinearity precondition imposed by GLM, Lasso, and Elastic Net models was tested using the Variance Inflation Factors (VIF) [28] calculated for HDL predictors. Table 3 shows the calculated VIF values for all predictors. Large VIF values exceeding 5-10 indicate an amount of collinearity that can be problematic to models like GLM. We notice that age, BMI, Blood Pressure—H, TG, LDL, glucose, and HbA1c levels present a high degree of multicollinearity.

A means of reducing multicollinearity is to remove some of the correlated predictors. VIFs were recalculated after omitting glucose levels (Table 3), which resulted in a slight decrease in VIFs. A supplementary omission of age (Table 3) decreased the VIFs of remaining parameters even more, but BMI (VIF = 43.58), Blood Pressure—H (VIF = 122.14), LDL (VIF = 70.97), and HbA1c (VIF = 73.82) still presented high VIFs. At this point, ChatGPT was asked to suggest further reductions. It analyzed the medical relevance of predictors and their potential multicollinearity, proposing the additional removal of Blood Pressure—H. The recalculation of VIFs showed improved values (Table 3) that are still insufficiently low to meet the no multicollinearity assumption. Based on the new results, ChatGPT envisaged a further reduction in predictors based on domain knowledge or relevance for HDLs. At the same time, it suggested accepting the multicollinearity. The multicollinearity of data can influence the accuracy of coefficient estimates and their interpretation in the case of the GLM model; therefore, the GLM model cannot be applied to the current data. The Lasso or Elastic Net models could handle multicollinearity to some degree, but we decided not to apply the Lasso or Elastic Net models either.

A common precondition for machine learning approaches (Random Forest, GBM, SVR, and Neural Networks) is to have no missing data in the training set for the variables under consideration, which is met for the analyzed dataset. In addition, SVR and neural networks require the evaluation of the dataset scale and decide whether a data normalization should be performed. The features have different scales. For instance, BMI ranges from 3.00 to 39.50, while TG ranges from 140 to 540. Therefore, the normalization or standardization of data is advisable before modeling.

#### 3.2.2. Data Modeling and Optimal Model Selection

The training set comprising data on diabetic cases was used for training Quantile Regression, Random Forest, and Gradient Boosting Machine (GBM) models. After normalization, the data were also used for training a Support Vector Regression (SVR) and a Neural Network model. The MSE values obtained for these models are 86.35 for Quantile Regression, 70.51 for Random Forest, 75.68 for GBM, 79.46 for SVR, and 73.49 for the Neural Network. The data are best modeled by the Random Forest approach, as it led to the lowest MSE. The following best-performing models were Neural Network and GBM.

Random Forest, GBM, and Neural Network models were further refined by hyperparameter tuning using the grid search approach. The resulting hyperparameters are presented in Table 4. Based on the MSE values obtained after hyperparameter tuning, the Random Forest model appears to perform best. Therefore, it was considered for evaluation and additional refinement using the data in the test set.

The Random Forest model with the tuned hyperparameters led to an MSE of 70.51 when applied to the test set. This value is close to the MSE obtained on the training set prior to hyperparameter tuning, which shows that the model is generalizing well to the new data.

#### 3.2.3. Data Relationships According to the Random Forest Model

The importance of features determined for the parameters in the context of the Random Forest model with tuned hyperparameters is 44.9% for TG, 23.3% for HbA1c, 20.7% for LDL, 5.0% for BMI, 3.9% for onset, and 2.1% for sex. This shows that TG presents the highest importance and subsequent contribution to the decision-making process. A mean importance is seen in the case of HbA1c and LDL, while BMI, onset, and sex present a low importance.

Partial dependence plots for TG, HbA1c, and LDL are presented in Figure 2. It appears that HDL levels decrease when TG levels increase or when HbA1c levels increase. There is only a slight decrease in HDL levels as LDL levels increase, but mostly, HDL levels are relatively constant over different LDL levels.

Several analyses were performed to gain a deeper understanding of the relationships between HDL levels and their predictors: distribution analysis, bivariate relationships, interaction effects, correlation analysis, and model residual analysis.

The distribution analysis consisted of visually observing the distribution of the primary predictors (TG, HbA1c, and LDL) over different HDL levels (Figure 3). HDL values were divided into quartiles, and the distribution of predictors was addressed relative to the quartiles. The distribution of TG shows that the parameter shifts to higher levels as HDL levels decrease. Less pronounced than in the case of TG, as HDL decreases, there is a slight shift toward higher HbA1c levels. LDL levels have a fairly consistent distribution across HDL levels.

The bivariate relationship analysis involved evaluating the relationship between HDL and its primary predictors using scatter plots (Figure 4). The TG versus HDL plot shows a negative relationship between the two properties. Similarly, but less pronounced, HbA1c also shows a negative relationship to HDL. As in the previous discussions, the values in the LDL versus HDL plots are widely dispersed, thus showing a less clear relationship.

The interaction effect analysis aimed to understand if the impact of one predictor on HDL levels is dependent on the level of another predictor. Given the unclear relation of LDL to HDL, the analysis focused on TG and HbA1C and their impact on HDL levels. More precisely, the analysis addressed the effect of TG on HDL for different levels of HbA1c (Figure 5). It appears that across all levels of HbA1c, mean HDL decreases as TG increases. This is more pronounced at low and medium levels of HbA1c and less pronounced at high levels of HbA1c. These observations suggest that TG levels have an overall negative effect on HDL, and the effect might be influenced by HbA1c levels.

The correlation analysis of the primary predictors and HDL revealed a negative correlation (r_TG-HDL_ = −0.69; r_HBA1c-HDL_ = −0.63; r_LDL-HDL_ = −0.75). The strongest correlation is seen between LDL and HDL, followed by TG and HbA1c.

The last analysis performed was the examination of the residuals calculated for the refined Random Forest model. This was performed using the residual plot in which predicted HDL values are plotted against the residuals (Figure 6). The scattering of residuals over different HDL values suggests that the model has a consistent variance across different levels of predicted HDL values (homoscedasticity). Also, in the absence of a clear pattern in residuals, the model appears to capture reasonably well the trends in the data.

### 3.3. Random Forest Models of All Variables

After validating the Random Forest model as best performing on our data, Random Forest models were trained to model each parameter (dependent variable) based on the others. MSE values obtained for these models are given in Table 5. The models for systolic and diastolic blood pressure, as well as for HbA1c, present the lowest MSE, supporting that these variables are best predicted using our data. The least predictable variables are TG and LDL.

The model for HbA1c was further considered for in-depth analysis. The model presented an MSE of 0.0656. The importance of the features for predicting HbA1c is 95.72% for Glucose, 1.71% for TG, 0.72% for HDL, 0.45% for sex, 0.36% for BMI, 0.32% for age, 0.30% for onset, 0.27% for LDL, 0.09% for Blood Pressure—H, and 0.007% for Blood Pressure—L. The model identified glucose levels as the most important for predicting HbA1c, while the other parameters appear to have small contributions. Blood pressure variables are the least important for the prediction, with a contribution close to zero.

Given that the relevance of glucose to HbA1c is well known, the model was trained when omitting glucose. The new model has an MSE of 0.2944, showing that the predictive power decreased when omitting glucose levels. This supports the importance of glucose levels for the levels of HbA1c but helps identify the following predictors of HbA1c. The calculated importance of features is 73.65% in the case of TG, 7.22% for HDL, 5.89% for age, 4.89% for BMI, 3.76% for LDL, 1.78% for onset, 1.15% for Blood Pressure—H, 1.13% for sex, 0.52% for sex, and 0.52% for Blood Pressure—L. This shows that TG levels became the main predictor, while the other parameters have a minimal influence on predicting HbA1c.

### 3.4. Modeling the Transition between Normal and High HbA1c Levels

The data on disease onset and HbA1c levels of diabetic patients were used to model the transition between normal and elevated HbA1c. The transition represents the event of interest that occurs when HbA1c levels exceed 6.5% [38]. The model aims to link the time since the initial diagnosis to the occurrence of the event. The patients who have not experienced the event are considered censored. The Kaplan–Meier survival curve [29] in Figure 7 presents the probability of maintaining HbA1c levels below the threshold. It shows an immediate decrease in HbA1c levels after initial diagnosis. The survival approaches zero after 6 years, indicating that almost all patients in the dataset experienced elevated HbA1c levels within the timeframe.

### 3.5. Discussions and Perspectives

In the present study, we performed a statistical analysis on data from diabetic and control patients harnessing the capabilities of the Code Interpreter Plugin of ChatGPT. The analysis has two main purposes. One was to derive novel information that can be of relevance to the medical field, and the other was to achieve a complex analysis with minimal computational and software resources. The present work is a cross-sectional study using routine analyses that are performed in Romanian clinics on diabetic patients (age, sex, BMI, systolic and diastolic blood pressure, TG, HDL, LDL, glucose and HbA1c levels, and disease onset) to understand the relationship between the measured parameters.

Using the available measurements on diabetic and control patients, we initially showed that there are strong and very strong correlations between these parameters in the case of diabetic patients, while there are no such correlations in the case of control patients. This is not surprising given the complex clinical picture of diabetes in which many parameters, including lipidic ones, are perturbed [5,6,7,38].

In the case of diabetic patients, the analysis showed that HDL is the only parameter found in an inverse strong and very strong correlation with the other parameters. Low HDL levels are considered a risk factor for cardiovascular disease [39]. HDL levels appear as a valuable predictor for cardiovascular events, especially since the patients using LDL-lowering agents still experienced such pathology [40,41]. HDL levels are very important for diabetic patients. Ikura et al. [42] also showed that lower HDL cholesterol levels are associated with extremity amputation and wound-related death in patients suffering from diabetic foot ulcers. Concerning the risk for cardiovascular diseases in diabetic patients, there is a U-shaped association between HDL and clinical outcomes due to a possible interaction with the glycemic status [43].

Given the relevance of HDL, here, we explored several models to understand the relationships and predict HDL values based on all the other parameters: age, BMI, systolic and diastolic blood pressure, TG, LDL, glucose, HbA1c levels, and disease onset. HDL modeling started with simple models like GLM, Lasso, or Elastic Net. The assumptions check showed that our data does not meet the multicollinearity assumption, suggesting that more complex models should be used to capture the relationship of HDL to the other parameters. The best-performing model for HDL prediction was the Random Forest model, which showed the high importance of TG, LDL, and HbA1c in influencing HDL levels. A more in-depth analysis of the model showed that HDL levels decrease as TG levels increase. Less pronounced was how HDL levels decrease as HbA1c levels increase, while LDL levels present a more constant distribution across all HDL levels. TG levels are the most important predictor of HDL. An interaction between HbA1c and TG levels was also seen; the impact of HbA1c on the HDL-TG dependence is more pronounced at lower to medium levels of HbA1c.

The research that we performed identified TG, HbA1c, and LDL as the main determinants of HDL by filtering a larger set of parameters. The results offer a more complex picture relative to studies focusing on pairs of parameters like HDL-TG, HDL-LDL [44], or HDL-HbA1c [45]. HDL, TG, and LDL are established risk factors for cardiovascular disease and are important therapeutic targets for treating them. Baliga et al. have discussed in [39] that there are therapeutic strategies that result in decreasing cardiovascular risk by elevating HDL levels. Other strategies for cardiovascular risk reduction involve the reduction of LDL levels [39] or TG levels [46]. Knowledge of the relationships between these parameters can be useful in the design of therapeutic strategies.

In addition to HDL modeling, we also explored the possibility of predicting other parameters using Random Forest models. Models with good prediction power were obtained for blood pressure—both systolic and diastolic and for HbA1c. The main predictor of HbA1c was glucose levels, but omitting glucose revealed that TG levels are the following main predictor. The survival analysis performed on HbA1c levels revealed that the transition from normal to elevated levels is expected to occur in the case of all patients within a 6-year timeframe after diagnosis. TG levels are the principal predictors of both HDL and HbA1c levels. At the same time, the levels of TG and LDL were the less predictable properties using Random Forest models. This suggests that an improved control of TG levels should have a positive impact on both glycosylated protein levels and HDL levels.

The study performed here can be easily extended by including other parameters. Being a cross-sectional study at this point, the follow-up of patients can provide additional insight into the time evolution of parameters, especially under specific therapies, and the diabetes complications that might arise.

## 4. Conclusions

We successfully used ChatGPT to analyze the complex relationships between twelve parameters measured on Romanian diabetic patients. In contrast to data from control patients, data from diabetic patients showed strong and very strong correlations. HDL is the only parameter in negative correlation with most of the others. Given the enhanced multicollinearity of data, HDL modeling required the usage of a more complicated model. The best-performing model was Random Forest. TG, LDL, and HbA1c resulted as the most important predictors of HDL, the increase in TG and HbA1c levels being associated with the decrease in HDL. The interaction analysis showed that the decrease in HDL when TG increased is seen at all HbA1c levels, especially at low and medium levels of HbA1c. The other parameters that can be predicted with Random Forest models are HbA1c levels and blood pressure, while the parameters that are difficult to predict are TG and LDL levels. The main predictor of HDL and the second most important predictor of HbA1c (after glucose levels) is the level of TG.

Such a study bringing insight into the dependence of HDL on other parameters is relevant for future therapeutic strategies that could consider tuning the predictors of HDL in order to have a positive impact on HDL. The study can be easily translated on a different parameter important for diabetes or for any other disease. The methodology that we proposed is easy to apply using the ChatGPT Code Interpreter. It does not require the installation of software and libraries, which can be difficult for non-specialists. Concerning the models that were obtained with ChatGPT, the Code Interpreter performed the modeling, offered explanations, identified limitations, and performed graphical data representations. Our work proves that ChatGPT can successfully assist the interpretation of scientific data, being a valuable tool for researchers with limited computational capabilities.

## Figures and Tables

**Figure 1 jpm-13-01466-f001:**
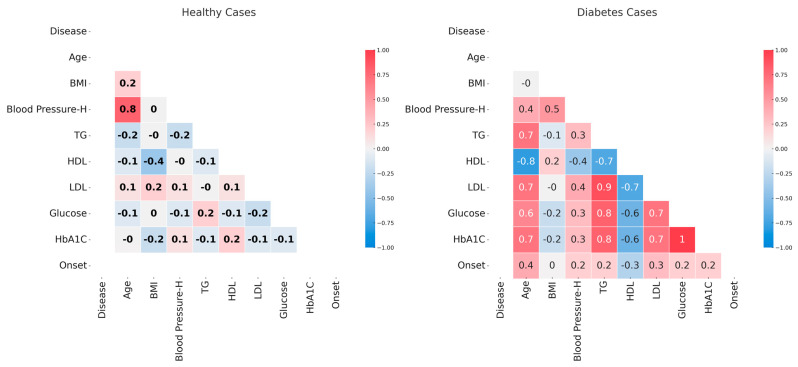
The correlation between analyzed parameters in the case of healthy, nondiabetic patients (**left plot**) and diabetic patients (**right plot**). Correlation coefficients are labeled. The color scheme varies from red—maximum inverse correlation (correlation coefficient of −1) to blue—maximum direct correlation (correlation coefficient of 1). The figure was generated in R [15].

**Figure 2 jpm-13-01466-f002:**
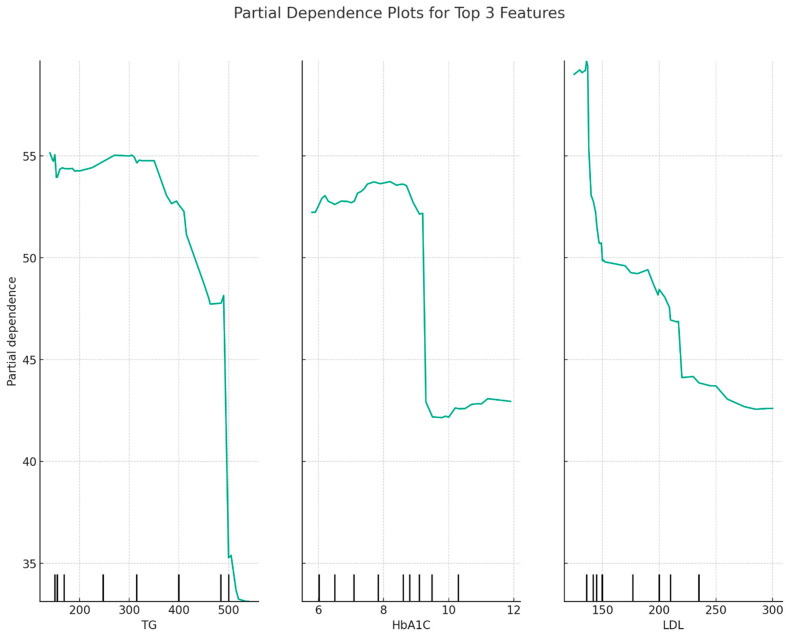
Partial dependence plots for TG, HbA1c, and LDL as revealed by the Random Forest model.

**Figure 3 jpm-13-01466-f003:**
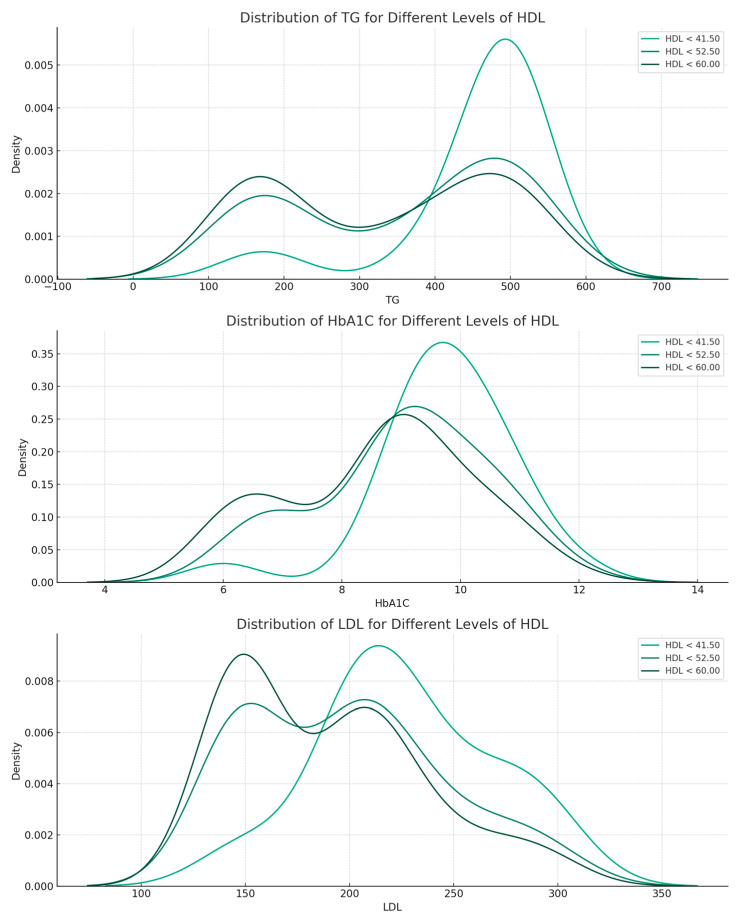
Distribution of TG, HbA1c, and LDL over different levels of HDL.

**Figure 4 jpm-13-01466-f004:**
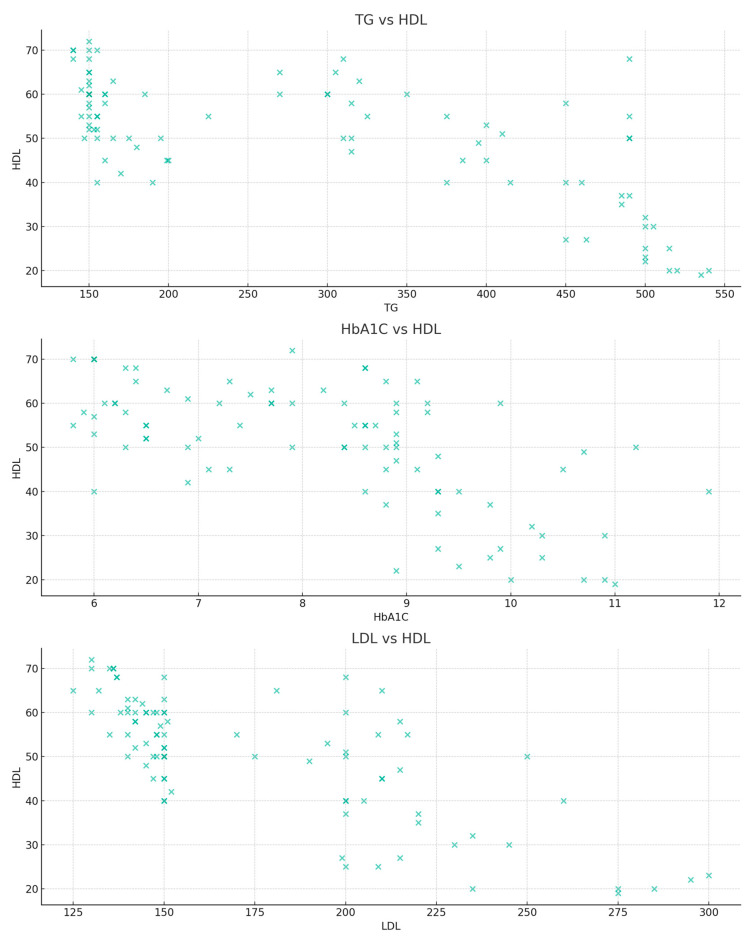
Scatter plots of the primary predictors—TG, HbA1c, and LDL vs. HDL values.

**Figure 5 jpm-13-01466-f005:**
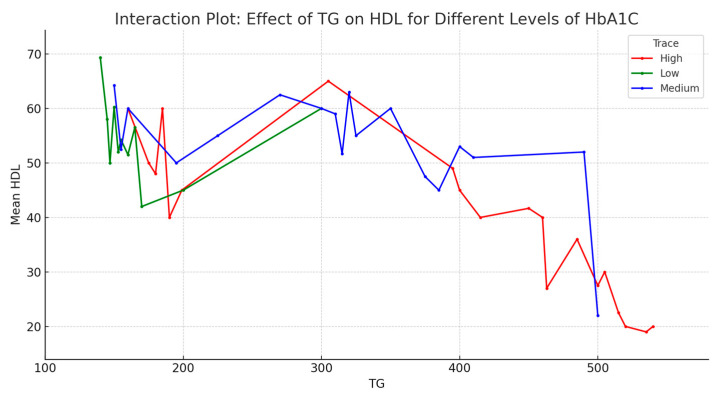
The effect of TG on HDL for different levels of HbA1c (interaction plot).

**Figure 6 jpm-13-01466-f006:**
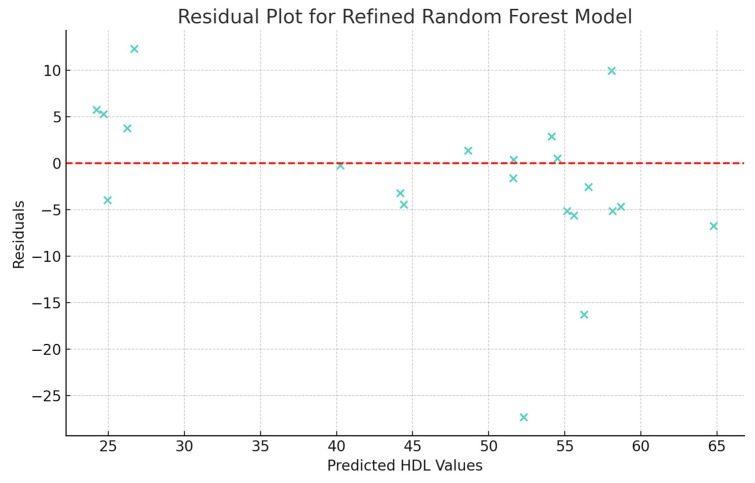
Residuals (differences between the actual and predicted HDL values) of the refined Random Forest model.

**Figure 7 jpm-13-01466-f007:**
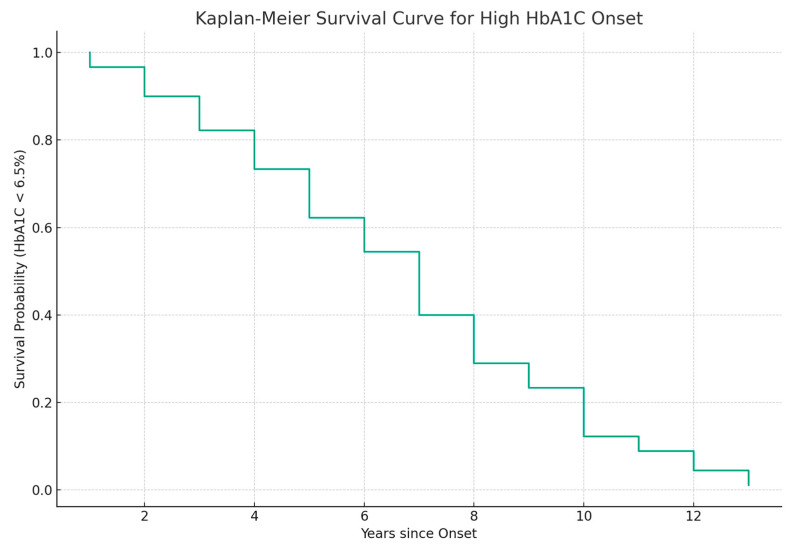
The Kaplan–Meier survival curve considering the transition from normal to elevated HbA1c levels (HbA1c ≥ 6.5%) as the event of interest.

**Table 1 jpm-13-01466-t001:** The analyzed dataset for the total of 160 diabetic and nondiabetic patients.

Variable	Type	Description	Non-Null Count	Range
Disease	Integer (0 or 1)	1 for Yes (the patient has diabetes) or 0 for No (the patient does not have diabetes),	160	NA
Sex	Object (Categorical—‘M’ or ‘F’)	Gender of the patient	160	NA
Age	Integer	Age of patient	160	35–70 years
BMI	Float	Body Mass Index of patient	160	3.0–40.0
Blood Pressure—H	Integer	Systolic blood pressure measurement	160	123–147 mmHg
Blood Pressure—L	Integer	Diastolic blood pressure measurement	160	81–91 mmHg
TG	Integer	Triglyceride level of patient	160	130–540 mg/dL
HDL	Integer	High-density lipoprotein level	160	19–72 mg/dL
LDL	Integer	Low-density lipoprotein level	160	60–300 mg/dL
Glucose	Integer	Blood glucose level	160	70–295 mg/dL
HbA1C	Float	Hemoglobin A1c level, indicating average blood sugar over the past 2–3 months	160	3.9–11.9%
Onset	Float	Time between the onset of diabetes and the moment of collecting data	110 (50 missing values)	1–13 years

**Table 2 jpm-13-01466-t002:** Preconditions of the applied regression models.

Model	Preconditions
Generalized Linear Model (GLM) [17]	-linear variation of predictors and outcome -independence of errors -homoscedasticity of errors -normality of error distribution -no or little multicollinearity
Lasso [18] and Elastic Net [19] Regressions	-evaluation of multicollinearity
Quantile Regression [20]	-linear variation of predictors and outcome -independence of errors
Random Forest [21,22], GBM [23], SVR, [24] and Neural Network [25]	-no missing data -data normalization for Neural Networks

**Table 3 jpm-13-01466-t003:** Variance Inflation Factors (VIFs) calculated over all predictors and when omitting one or more parameters, as noted in the head of the table.

Variable	VIF—All Variables	VIF after Omitting ‘Glucose’	VIF after Omitting ‘Glucose’ and ‘Age’	VIF after Omitting ‘Glucose’, ‘Age’, and ‘Blood Pressure—H’
Sex	2.09	2.09	2.09	2.07
Age	117.24	115.19	-	-
BMI	43.82	43.60	43.58	22.70
Blood Pressure—H	189.24	155.07	122.14	-
TG	26.99	26.40	24.00	20.77
LDL	71.97	71.17	70.97	62.39
Glucose	509.68	-	-	-
HbA1c	825.05	79.00	73.82	46.19
Onset	8.05	7.92	6.78	6.76

**Table 4 jpm-13-01466-t004:** The hyperparameters of Random Forest, GBM, and Neural Network models tuned by grid search and the resulting best MSE for each model.

Model	Hyperparameter	Significance of Hyperparameter	Value	Best MSE
Random Forest	n_estimators	Number of trees in the forest	100	55.48
	max_depth	Maximum depth of the tree	None	
	min_samples_split	Minimum samples required to split an internal node	2	
Neural Network (MLPRegressor)	hidden_layer_sizes	Number of neurons in hidden layers	100	74.46
	activation	Activation function for the hidden layer	tanh	
	alpha	L2 penalty (regularization term) parameter	0.0001	
GBM	n_estimators	Number of boosting stages to run	50	62.05
	learning_rate	Step size shrinkage used to prevent overfitting	0.1	
	max_depth	Maximum depth of the individual regression estimators	3	

**Table 5 jpm-13-01466-t005:** MSE calculated for the Random Forest models trained by considering each parameter as dependent variable and the remaining as independent variables.

Dependent Variable	Mean Squared Error (MSE)
Blood Pressure—L	0.0163
HbA1c	0.0753
Blood Pressure—H	0.1105
Sex	0.1498
Onset	12.7679
Age	12.9156
BMI	14.8431
HDL	53.59.98
Glucose	53.9048
LDL	421.2393
TG	745.7605

## Data Availability

The data presented in this study are available on reasonable request from the corresponding author.

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
