# Peer review of "Harnessing Code Interpreters for Enhanced Predictive Modeling: A Case Study on High-Density Lipoprotein Level Estimation in Romanian Diabetic Patients"

_jpm, 2023, doi:10.3390/jpm13101466_

Round 1
Reviewer 1 Report
The study provides an interesting method to assess the relationship between some specific laboratory and clinical parameters in diabetes and provide mathematical models of prediction of HDL levels. However, most of the findings are well-known, and the results of this study are not innovative. In addition, the readability of the text is difficult in some parts of the manuscript. For example, in the presentation of the results, the authors describe their findings along with a discussion.
It needs to be clarified whether the variables you included in the analyses were collected cross-sectionally, or you calculated mean/median values through a (pre-specified?) observation period, or, instead, you considered the trend across a (pre-specified?) observation period. The comment is a crucial point, as arterial pressure and lipids are common targets of T2D treatment, and they are susceptible to change considerably over the follow-up under the effect of specific therapies. From the latter viewpoint, the predicting model could be more useful as, based upon your experience in managing patients with T2D, you could predict therapeutic responses of every single patient (e.g., 1-year, 5-year, 10-year responses) in terms of any specific end-points (HDLc, LDLc, arterial pressure, HbA1c) for clustering treatment strategies more efficiently.
According to what was mentioned above, as an alternative study design, the paper would be improved by working on models to predict the risk of adverse outcomes in T2D starting from well-known laboratory and clinical parameters.
As another limitation, why didn't you consider HbA1c?
Check the text for typing errors and editing.
Author Response
Reviewer 1:
The study provides an interesting method to assess the relationship between some specific laboratory and clinical parameters in diabetes and provide mathematical models of prediction of HDL levels. However, most of the findings are well-known, and the results of this study are not innovative. In addition, the readability of the text is difficult in some parts of the manuscript. For example, in the presentation of the results, the authors describe their findings along with a discussion.
It needs to be clarified whether the variables you included in the analyses were collected cross-sectionally, or you calculated mean/median values through a (pre-specified?) observation period, or, instead, you considered the trend across a (pre-specified?) observation period. The comment is a crucial point, as arterial pressure and lipids are common targets of T2D treatment, and they are susceptible to change considerably over the follow-up under the effect of specific therapies. From the latter viewpoint, the predicting model could be more useful as, based upon your experience in managing patients with T2D, you could predict therapeutic responses of every single patient (e.g., 1-year, 5-year, 10-year responses) in terms of any specific end-points (HDLc, LDLc, arterial pressure, HbA1c) for clustering treatment strategies more efficiently.
According to what was mentioned above, as an alternative study design, the paper would be improved by working on models to predict the risk of adverse outcomes in T2D starting from well-known laboratory and clinical parameters.
As another limitation, why didn't you consider HbA1c?
Check the text for typing errors and editing.
Response:
We thank the reviewer for evaluating our manuscript and for the valuable suggestions. These were implemented as follows:
- HbA1c levels were included in the study, along with disease onset.
- The statistical analysis of data was changed to the changes in the dataset.
- We clearly mentioned in the manuscript that the data is cross-sectional.
- We separated the results from the discussion.
- The analysis focused on finding the associations based on many parameters and revealed some known correlations like those between HDL levels, TG and LDL levels. Still, we consider that the novelty of our work consists in the data analysis method using ChatGPT and the Random Forest model. In the context of the model, we discuss not only the correlations between HDL, TG, LDL and HbA1c, but also interactions between the parameters like TG and HbA1c.
Reviewer 2 Report
Dear Editor in Chief Professor Rizzieri,
Thanks for giving me the opportunity to review this interesting manuscript entitled “Harnessing Code Interpreters for Enhanced Predictive Modeling: A Case Study on HDL Level Estimation in Romanian Diabetic Patients”. The article is interesting but has some design, methodological and results interpretation pitfalls. Attached below my comments that I hope would add to the manuscript.
Best regards,
Comments to the Authors,
I read with great interest the manuscript entitled “Harnessing Code Interpreters for Enhanced Predictive Modeling: A Case Study on HDL Level Estimation in Romanian Diabetic Patients”. It’s an interesting artificial intelligence based study.
Major comments:
Methodology:
It would have been better if the authors mentioned the age and diabetes duration of the patients. Moreover, adding controls would help in comparison if these relations are specific to patients with T2DM or they should be generalized.
Was ethical committee approval taken. If taken, would you please add its number.
Were the lipid profile and blood glucose collected and analyzed or they were taken from the patients records.
It would be nice to add data about the presence or absence of diabetes complications and HbA1C.
Multivariate logistic regression could give an idea about the factors independently correlated with overweight and obesity in these children.
Results:
Please mention the r values in figure 1.
It would be nice to add the abbreviations for table 3.
Discussion: The discussion is rather narrative It’s better to discuss the findings with the related studies one by one.
Conclusion:
The conclusion should not repeat the results but rather summarize the main outcome and its clinical implications.
Minor comments:
It is preferable to avoid using the term we, our study.
English editing is required
Author Response
Reviewer 2:
Methodology:
It would have been better if the authors mentioned the age and diabetes duration of the patients. Moreover, adding controls would help in comparison if these relations are specific to patients with T2DM or they should be generalized.
Was ethical committee approval taken. If taken, would you please add its number.
Were the lipid profile and blood glucose collected and analyzed or they were taken from the patients records.
It would be nice to add data about the presence or absence of diabetes complications and HbA1C.
Multivariate logistic regression could give an idea about the factors independently correlated with overweight and obesity in these children.
Results:
Please mention the r values in figure 1.
It would be nice to add the abbreviations for table 3.
Discussion: The discussion is rather narrative It’s better to discuss the findings with the related studies one by one.
Conclusion:
The conclusion should not repeat the results but rather summarize the main outcome and its clinical implications.
Minor comments:
It is preferable to avoid using the term we, our study.
English editing is required
Response:
We thank the reviewer for evaluating our manuscript and for the valuable suggestions. These were implemented as follows:
-The ranges of all analyzed parameters are presented
-We included the onset of the disease in the study, along HbA1c levels
-We included control patients in the initial assessment of correlations between the measurements
-The data was generated for the study and we mentioned the time period of recruiting patients
-Correlation coefficients were labeled in Figure 1
-Table 3 was excluded from the manuscript; the whole results section was rewritten according to the new statistical analysis resulting from including new parameters
-The discussions were written separately
-The conclusion was rewritten
-“we” was avoided, thank you for this suggestion as well. English was improved, however, a full english editing will be performed thoroughly by a native if the manuscript passes the peer review process.
Round 2
Reviewer 1 Report
The general consideration of the manuscript remains the same. However, the manuscript has slightly improved in organization and data presentation.
Check for typos.
Reviewer 2 Report
All previous comments were answered.
The manuscript is suitable for publication in the current form.